# Complete Chloroplast Genomes of Three Medicinal *Alpinia* Species: Genome Organization, Comparative Analyses and Phylogenetic Relationships in Family Zingiberaceae

**DOI:** 10.3390/plants9020286

**Published:** 2020-02-24

**Authors:** Dong-Mei Li, Gen-Fa Zhu, Ye-Chun Xu, Yuan-Jun Ye, Jin-Mei Liu

**Affiliations:** Guangdong Key Lab of Ornamental Plant Germplasm Innovation and Utilization, Environmental Horticulture Research Institute, Guangdong Academy of Agricultural Sciences, Guangzhou 510640, China; tahua@126.com (Y.-C.X.); yeyuanjun199010@126.com (Y.-J.Y.);

**Keywords:** *Alpinia katsumadai*, *Alpinia oxyphylla*, *Alpinia pumila*, chloroplast genome, comparative analysis, single nucleotide polymorphisms, indels, phylogenetic relationship

## Abstract

*Alpinia katsumadai* (*A. katsumadai*), *Alpinia oxyphylla* (*A. oxyphylla*) and *Alpinia pumila* (*A. pumila*), which belong to the family Zingiberaceae, exhibit multiple medicinal properties. The chloroplast genome of a non-model plant provides valuable information for species identification and phylogenetic analysis. Here, we sequenced three complete chloroplast genomes of *A. katsumadai*, *A. oxyphylla* sampled from Guangdong and *A. pumila*, and analyzed the published chloroplast genomes of *Alpinia zerumbet* (*A. zerumbet*) and *A. oxyphylla* sampled from Hainan to retrieve useful chloroplast molecular resources for *Alpinia*. The five *Alpinia* chloroplast genomes possessed typical quadripartite structures comprising of a large single copy (LSC, 87,248–87,667 bp), a small single copy (SSC, 15,306–18,295 bp) and a pair of inverted repeats (IR, 26,917–29,707 bp). They had similar gene contents, gene orders and GC contents, but were slightly different in the numbers of small sequence repeats (SSRs) and long repeats. Interestingly, fifteen highly divergent regions (*rpl36*, *ycf1*, *rps15*, *rpl22*, *infA*, *psbT-psbN*, *accD-psaI*, *petD-rpoA*, *psaC-ndhE*, *ccsA-ndhD*, *ndhF-rpl32*, *rps11-rpl36*, *infA-rps8*, *psbC-psbZ*, and *rpl32-ccsA*), which could be suitable for species identification and phylogenetic studies, were detected in the *Alpinia* chloroplast genomes. Comparative analyses among the five chloroplast genomes indicated that 1891 mutational events, including 304 single nucleotide polymorphisms (SNPs) and 118 insertion/deletions (indels) between *A. pumila* and *A. katsumadai*, 367 SNPs and 122 indels between *A. pumila* and *A. oxyphylla* sampled from Guangdong, 331 SNPs and 115 indels between *A. pumila* and *A. zerumbet*, 371 SNPs and 120 indels between *A. pumila* and *A. oxyphylla* sampled from Hainan, and 20 SNPs and 23 indels between the two accessions of *A. oxyphylla*, were accurately located. Additionally, phylogenetic relationships based on SNP matrix among 28 whole chloroplast genomes showed that *Alpinia* was a sister branch to *Amomum* in the family Zingiberaceae, and that the five *Alpinia* accessions were divided into three groups, one including *A. pumila*, another including *A. zerumbet* and *A. katsumadai*, and the other including two accessions of *A. oxyphylla*. In conclusion, the complete chloroplast genomes of the three medicinal *Alpinia* species in this study provided valuable genomic resources for further phylogeny and species identification in the family Zingiberaceae.

## 1. Introduction

*Alpinia* Roxb. is the biggest genus in the family Zingiberaceae, which includes more than 230 species of perennial herb [1,2,3]. Almost all of the *Alpinia* species center in Southeast Asia, but western outposts of distribution occur in India and extend south into Australia and the islands of the South Pacific [1,2,3]. There are 54 species in China and most of them possess multiple medicinal properties; these species include *Alpinia katsumadai* K. Schum., *Alpinia oxyphylla* Miq., *Alpinia pumila* Hook. f., and others [1,2,4]. Of these species, the dried seeds of *A. katsumadai* and *A. oxyphylla*, and dried rhizomes of *A. pumila* can be used as traditional Chinese medicine and folk medicine, respectively [1,2,4]. *A. katsumadai* has synonyms *Alpinia hainanensis*, *Alpinia henryi* var. *densihispida*, *Alpinia kainantensis*, and *Alpinia katsumadae* [1,3]. *A. katsumadai* is mainly distributed in forests of Southern China (Guangdong, Guangxi, and Hainan provinces) as well as south into neighboring Vietnam [1,3]. *A. pumila* is native to shady, humid mountain valleys in Guangdong, Guangxi, Hunan and Yunnan provinces at altitudes of 500–1100 m [1,3]. *A. oxyphylla* is widely cultivated in Fujian, Guangdong, Guangxi, Hainan, and Yunnan provinces of China [1]. Due to the medicinal value of these three *Alpinia* species, numerous studies have focused on identifying the effective chemical constituents of these plants, which have many pharmacological effects—for instance, anti-bacterial activity, anti-inflammatory, anti-neoplastic and alzheimer’s disease [4,5,6,7,8,9,10]. 

Few vegetative characteristics are observed when *Alpinia* species do not produce bright-colored flowers or fruits, as a result causing difficulty in identifying the species of *Alpinia* [1,2,3,11]. This genus has also been molecularly studied [12,13]. Results have showed a sufficient resolution among 12 *Alpinia* species (*Alpinia conchigera*, *Alpinia galanga*, *Alpinia elegans*, *Alpinia luteocarpa*, *Alpinia vittata*, *Alpinia blepharocalyx*, *Alpinia intermedia*, *A. pumila*, *Alpinia calcarata*, *Alpinia officinarum*, *Alpinia foxworthyi* and *Alpinia carolinensis*) and their phylogenetic relationships using nuclear internal transcribed spacer (*ITS*) and chloroplast genome *matK* regions [12]. In a recent study, phylogenetic relationships between *Alpinia zerumbet* and *A. oxyphylla* sampled from Hainan were evaluated by the whole chloroplast genomes [13]. These publicly available two chloroplast genomes of *Alpinia* are insufficient to resolve morphological differences at interspecific and intraspecific levels, especially failing to distinguish among three medicinal species, namely, *A. katsumadai*, *A. pumila* and *A. oxyphylla*, and different sources of *A. oxyphylla* [12,13], because *A. katsumadai* and different sources of *A. oxyphylla* were not included and their phylogenetic positions were still unresolved. Therefore, we attempted to report the complete chloroplast genomes of *A. katsumadai*, *A. pumila* and *A. oxyphylla* sampled from Guangdong, as well as to explore their phylogenetic relationships. 

The chloroplast is an important organelle that can conduct photosynthesis and produce essential energy in green plants [14,15]. It has its own independent genome, which comprises a closed circular DNA molecule [14,15]. Typical chloroplast genomes of angiosperms have a generally conserved quadripartite circular structure, which consists of a large single copy (LSC) region, a small single copy (SSC) region, and two copies of inverted repeats (IRs) [16,17,18]. With the rapid development of high-throughput sequencing technologies, such as Illumina and PacBio sequencing platforms, it is now convenient to acquire complete chloroplast genome sequences [19,20,21,22,23,24,25,26,27,28,29,30,31,32,33,34]. The chloroplast genomes have been widely utilized for species identification, the reconstruction of phylogenetic relationships, the resolution of origin problems, and the development of molecular markers [19,20,21,22,23,24,25,26,27,28,29,30]. 

In the current study, we firstly sequenced the complete chloroplast genomes of *A. katsumadai*, *A. pumila* and *A. oxyphylla* sampled from Guangdong, using Illumina and PacBio sequencing platforms, respectively. Then, we compared the resulting chloroplast genomes with the published chloroplast genomes of *A. zerumbet* (JX088668) and *A. oxyphylla* sampled from Hainan (NC_035895) [13]. Our main objectives were as shown below: (1) to explore the molecular structures of three chloroplast genomes of *Alpinia* in this study; (2) to examine the variations of simple sequence repeats (SSRs) and long repeats among the five *Alpinia* chloroplast genomes; (3) to discover highly divergent regions that could be used as specific DNA markers for *Alpinia*; (4) to deeply understand the interspecific and intraspecific variations among the five *Alpinia* chloroplast genomes; (5) to reveal phylogenetic relationships among *A. katsumadai*, *A. pumila* and *A. oxyphylla* in family Zingiberaceae. 

## 2. Results and Discussion

### 2.1. The Chloroplast Genome Features of Alpinia Species

All the three species of *Alpinia* we sequenced had a typical quadripartite structure, with a single circular molecule ranged from 161,410 bp (*A. oxyphylla* sampled from Guangdong) to 162,387 bp (*A. katsumadai*) in length. They had four junction regions: a separate LSC region of 87,261–87,667 bp, an SSC region of 15,306–16,180 bp, and a pair of IRs (IRa and IRb) each 28,964–29,707 bp (Figure 1 and Table 1 and Appendix A). The size of the *A. katsumadai* chloroplast genome (162,387 bp) was the largest among the three sequenced *Alpinia* species, with 977 bp longer than that of *A. oxyphylla* sampled from Guangdong and 467 bp longer than that of *A. pumila*. The GC content of the three species chloroplast genomes varied slightly from 36.15% to 36.17% (Table 1 and Appendix A). The AT content at the third codon position (71.35%–71.37%) was higher than that at the first (55.30%–55.44%) and second (62.50%–62.59%) positions in the protein-coding genes of these three *Alpinia* species (Appendix A). Additionally, the AT content was the highest (70.18–70.38%) in the SSC region, the lowest (50.48%–50.79%) in the IR regions, and moderate (66.14%–66.18%) in the LSC region (Appendix A). These genomic structures were consistent with most other published chloroplast genomes of family Zingiberaceae, such as two *Kaempferia* species [23], three *Amomum* species [24], *Zingiber officinale* [25], *Stahlianthus involucratus* [31], *Hedychium coronarium* [32] and *Curcuma longa* [33]. 

A total of 133 predicted genes, consisting of 87 protein-coding genes, 38 tRNA genes, and 8 rRNA genes, were detected in each chloroplast genome of our sequenced three species of *Alpinia* (Table 1, Table 2 and Appendix A). Both of the chloroplast genomes of *A. zerumbet* and *A. oxyphylla* sampled from Hainan contained 132 genes (Table 1). As shown in Table 1, the *A. zerumbet* chloroplast genome had the highest GC content (36.27%), while the *A. katsumadai* chloroplast genome had the lowest GC content (36.15%). The length of the *A. katsumadai* chloroplast genome was the longest and the *A. zerumbet* chloroplast genome (159,773 bp) was the shortest. Interestingly, the SSC region of *A. katsumadai* (15,306 bp) was the shortest, whereas the SSC region of the *A. zerumbet* chloroplast genome (18,295 bp) was the longest (Table 1). The complete chloroplast genome of *A. oxyphylla* sampled from Guangdong was 59 bp longer than that of *A. oxyphylla* sampled from Hainan (Table 1). The lengths of the IR regions of the five chloroplast genomes, ranging from 26,917 to 29,707 bp (Table 1), were shorter than those of the three species of *Amomum*, which varied from 29,820 to 29,959 bp [24]. In addition, 86 protein-coding genes were identified in the *A. zerumbet* chloroplast genome, and 87 were identified in the other four accessions. Eight conserved rRNAs were identified in every chloroplast genome. The chloroplast genome of *A. oxyphylla* sampled from Hainan encoded 37 types of tRNAs, and the other four accessions encoded 38 (Table 1). 

A total of 20 genes were duplicated in the IR regions, including eight protein-coding genes (*ndhB*, *rpl2*, *rpl23*, *rps7*, *rps12*, *rps19*, *ycf1*, *ycf2*), eight tRNA genes (*trnH-GUG*, *trnI-CAU*, *trnL-CAA*, *trnV-GAC*, *trnI-GAU*, *trnA-UGC*, *trnR-ACG*, *trnN-GUU*), and all four rRNAs (*rrn4.5*, *rrn5*, *rrn16* and *rrn23*) (Figure 1 and Appendix A). Sixteen genes (*trnA-UGC*, *trnI-GAU*, *trnG-GCC*, *trnK-UUU*, *trnL-UAA*, *trnV-UAC*, *atpF*, *ndhA*, *ndhB*, *rpoC1*, *petB*, *petD*, *rpl2*, *rpl16*, *rps12* and *rps16*) contained one intron, while *ycf3* and *clpP* each contained two introns (Appendix A). Among the 18 intron-containing genes, four genes (*trnA-UAC*, *trnI-GAU*, *rpl2* and *ndhB*) occurred in the both IRs, 12 genes (*trnG-GCC*, *trnK-UUU*, *trnL-UAA*, *trnV-UAC*, *atpF*, *rpoC1*, *petB*, *petD*, *rpl16*, *rps16*, *ycf3* and *clpP*) were distributed in the LSC, one gene (*ndhA*) was in the SSC, and one gene (*rps12*) was located its first exon in the LSC and the other two exons in both IRs (Figure 1 and Appendix A). In addition, the three *Alpinia* species had the longest introns of *trnK-UUU* (2,653 bp, 2654 bp, 2626 bp, respectively), all of which were included in the coding region of *matK* (Appendix A).

### 2.2. Codon Usage and Predicted RNA Editing Sites Analyses

All the protein-coding genes were composed of 27,427–27,669 codons in the three chloroplast genomes of *Alpinia* species. Of the 27,427–27,669 codons, leucine (Leu) was the most abundant amino acid, with a frequency of 10.34%–10.38%, then isoleucine (Ile) with a frequency of 8.74%–8.80%, while cysteine (Cys) was the least common one with a proportion of 1.12%–1.13% (Figure 2 and Appendix A). This phenomenon was consistent with other land plants’ chloroplast genomes, such as *Z. officinale* [25], *Ailanthus altissima* [35], *Lycium chinense* [36], *Symplocarpus renifolius* [37], and *Epipremnum aureum* [38]. Due to the value of relative synonymous codon usage (RSCU) >1, thirty codons showed the codon usage bias in the chloroplast genes of all the three *Alpinia* species (Appendix A). Interestingly, out of the above 30 codons, twenty-nine codons were A/T-ending codons. Conversely, the C/G-ending codons had RSCU values of less than one, indicating that they were less common in the chloroplast genes of the three *Alpinia* species. Stop codon usage was found to be biased toward TAA (RSCU > 1.00). Two amino acids, methionine (Met) and tryptophan (Trp), showed no codon bias both with RSCU values of 1.00 (Appendix A). 

A total of 56 editing sites in 22 protein-coding genes were identified in *A. katsumadai*, while similar numbers were found in *A. pumila* (54 sites) and *A. oxyphylla* sampled from Guangdong (55 sites) (Appendix A). In the three *Alpinia* species chloroplast genomes we sequenced, the *ndhB* gene had the highest number of potential editing sites (11), followed by the *accD* gene (5). Similar to other species, such as *Kaempferia galanga* [23], *Kaempferia elegans* [23] and *Forsythia suspense* [39], the *ndhB* gene contained the largest number of editing sites. All these editing sites were C-to-T transitions and occurred at the first or second positions of the codons. Interestingly, most conversions at the codon positions changed from serine (S) to leucine (L) and most RNA editing sites led to amino acid changes for hydrophobic products, such as leucine, isoleucine, tryptophan, tyrosine, valine, methionine, and phenylalanine (Appendix A). Similar RNA editing features have already been revealed by previous observations [23,39]. 

### 2.3. SSRs and Long Repeats Analyses

SSRs were widely dispersed in chloroplast genomes, and have been extensively used in population genetics and molecular phylogenetic researches [40,41]. In this study, 1205 SSRs were identified in five *Alpinia* chloroplast genomes, including two accessions of *A. oxyphylla*, *A. katsumadai*, *A. zerumbet*, and *A. pumila* (Figure 3 and Appendix A). The most abundant were mononucleotide repeats, located in non-coding regions, and contributed to AT richness (Figure 3). These results are consistent with most reported angiosperms [23,24,25,26,28,29,30]. The total number of SSRs was 237 in *A. katsumadai*, 244 in *A. oxyphylla* sampled from Guangdong, 247 in *A. pumila*, 236 in *A. zerumbet*, and 241 in *A. oxyphylla* sampled from Hainan (Figure 3A). In the genomic structure of five chloroplast genomes, the non-coding region had the most abundant SSRs, whereas the coding region had the least SSRs (Figure 3A). The majority of SSRs were located in the LSC regions rather than in the SSC and IR regions of the five chloroplast genomes (Figure 3B). Mono-, di-, tri- and tetranucleotide SSRs were all detected in the five chloroplast genomes (Figure 3C). Additionally, pentanucleotide SSRs were found in two accessions of *A. oxyphylla* and *A. pumila*, respectively. Mononucleotide repeats were the largest in a number of these SSRs, with 77.63% and 78.54% found in *A. katsumadai* and *A. pumila*, respectively (Figure 3C). A/T repeats were the most common of mononucleotides (70.90%–76.69%), while AT/AT repeats were the majority of dinucleotide repeat sequences (92.30%–94.59%). Interestingly, our results show that AAAT/ATTT repeats were the third abundant SSR types in the five chloroplast genomes (2.96%–3.79%) (Figure 3D).

Additionally, we detected long repeats in five chloroplast genomes using REPuter, including forward, complement, reverse, and palindromic repeats. A total of 225 unique long repeats were found from the five chloroplast genomes. In detail, there were 44 (16 forward, 26 palindrome, two reverse), 35 (10 forward, 24 palindrome, one complement), 50 (18 forward, 27 palindrome, three reverse, two complement), 46 (19 forward, 26 palindrome, one reverse) and 50 (17 forward, 29 palindrome, two reverse, two complement) long repeats in *A. katsumadai*, *A. pumila*, *A. oxyphylla* sampled from Guangdong, *A. zerumbet* and *A. oxyphylla* sampled from Hainan, respectively (Figure 4A and Appendix A). Interestingly, there were no complement repeats in *A. katsumadai*, similar to *A. zerumbet*. With 29 palindrome repeats, *A. oxyphylla* sampled from Hainan contained the highest number of palindrome repeats, while *A. zerumbet* contained the highest number of forward repeats at 19, and *A. oxyphylla* sampled from Guangdong contained three reverse repeats, the highest among the compared genomes (Figure 4B–D). The palindrome, forward and reverse repeats with 30–60 bp were found to be the most common in the five chloroplast genomes (Figure 4B–D). Moreover, almost all of the lengths of the reverse repeats were less than 60 bp in the five chloroplast genomes (Figure 4D).

### 2.4. IR Contraction and Expansion Analyses

The contractions and expansions at the borders of IR regions were common evolutionary events and may cause size variations of chloroplast genomes [13,23,24,25,26,27,28,29,30]. We compared the IR-SC boundaries information of the five *Alpinia* chloroplast genomes (Figure 5). The genes *rpl22*, *rps19*, pseudogene *ycf1* (*Ψycf1*), *ndhF*, *ycf1* and *psbA* were present at the junction of the LSC/IRa, IRa/SSC, SSC/IRb, and IRb/LSC borders. As shown in Figure 5, the *rpl22*-*rps19* genes were located in the junctions of the LSC/IRa regions of the five chloroplast genomes. There were 47 bp between *rpl22* and the LSC/IRa borders, meanwhile, 129–130 bp between the *rps19* and the other LSC/IRa junctions. 

The *Ψycf1*-*ndhF* genes were located at the junctions of the IRa/SSC regions for all five chloroplast genomes; IRa/SSC borders of three species (*A. zerumbet*, *A. oxyphylla* sampled from Hainan and *A. katsumadai*) were all situated just adjacent to the end of *Ψycf1*; *Ψycf1* expanded into the SSC regions for 6 bp in *A.oxyphylla* sampled from Guangdong and 22 bp in *A. pumila*, respectively (Figure 5). In the three species (*A. zerumbet*, *A. oxyphylla* sampled from Hainan and *A. katsumadai*), the distances between *ndhF* and IRa/SSC border were 251 bp, 42 bp, and 2 bp, respectively. There were 24 bp and 42 bp between *ndhF* and *Ψycf1* border in *A. pumila* and *A. oxyphylla* sampled from Guangdong, respectively (Figure 5). 

The SSC/IRb junction was situated in the *ycf1* coding region, which crossed into the IRb region in all five chloroplast genomes. However, the length of *ycf1* in the IRb region varied among the five chloroplast genomes from 1048 bp to 3844 bp, which indicated the dynamic variation of the SSC/IRb junctions (Figure 5). 

The *rps19*-*psbA* genes were located in the junctions of the IRb/LSC regions in five chloroplast genomes (Figure 5). In the five chloroplast genomes, the distances between *rps19* and IRb/LSC border were 130 bp, 129 bp, 130 bp, 130 bp and 130 bp, respectively. For all five chloroplast genomes, 93–109 bp distance was found between *psbA* gene and the IRb/LSC border (Figure 5). Taken together, these data indicated that the contractions and expansions of the IR regions exhibited relatively stable patterns within genus *Alpinia*, with slight variations. 

### 2.5. Divergence Hotspot Regions Analyses

The whole chloroplast genome sequence of *A. pumila* (MK262731) was compared to those of *A.katsumadai* (MK262728), *A.oxyphylla* sampled from Guangdong (MK262729), *A. zerumbet* (JX088668) and *A.oxyphylla* sampled from Hainan (NC_035895) using the mVISTA program (Figure 6). The comparison showed that the two IR regions were less divergent than the LSC and SSC regions and that lower divergence was found in coding regions than in non-coding regions (Figure 6). In the coding regions, most genes were relatively conserved, except for *matK*, *petB*, *rps15*, *rpl22* and *ycf1*. The highly divergent regions were mainly located in the intergenic regions, such as *trnH*-*psbA*, *atpI-atpH*, *psbM-petN*, *trnE*-*psbD*, *psbC-psbZ*, *accD-psaI*, *psbT-psbN*, *petD-rpoA*, *ccsA-ndhD*, *ndhF-rpl32*, *rpl32*-*ccsA*, and *psaC-ndhE* (Figure 6). 

Furthermore, the five *Alpinia* accessions were observed to have highly variable regions in their chloroplast genomes by sliding window analysis using software DnaSP (Figure 7). Of the 64 protein-coding regions (CDS), nucleotide diversity (Pi) for each locus ranged from 0.0006 (*ycf2*) to 0.00877 (*rpl36*) and had the average value of 0.00242. Thereby, five regions (*rpl36*, *ycf1*, *rps15*, *rpl22* and *infA* genes) located at the LSC and SSC regions had remarkably high values (Pi > 0.005; Figure 7A). For the 72 non-coding regions, Pi values ranged from 0.00056 (*rpl20*-*rps12*) to 0.04918 (*psbT*-*psbN*) and had the average of 0.00746. Ten of these regions also showed remarkably high values (Pi > 0.012), including *psbT-psbN*, *accD-psaI*, *petD-rpoA*, *psaC-ndhE*, *ccsA-ndhD*, *ndhF-rpl32*, *rps11-rpl36*, *infA-rps8*, *psbC-psbZ*, and *rpl32-ccsA* (Figure 7B). These results also prove that the IR regions were more conserved than the LSC and SSC regions, and the average value of Pi in the non-coding regions was more than three times as much as in the coding regions. Among these regions, *ycf1*, *rps15*, *rpl22*, *infA*, *psbT-psbN*, *petD-rpoA*, *psaC-ndhE*, *ccsA-ndhD*, *ndhF-rpl32*, and *rpl32-ccsA* have also been reported as highly variable regions in other plant species, such as *Kaempferia* species [23], *Aristolochis* species [26], orchid species [42], Lythraceae species [43], *Quercus* species [44], *Lilium* species [45], *Croomia* species [46], *Stemona* species [46], and *Eucommia* species [47]. The *ndhF-rpl32* and *ccsA-ndhD* regions had been used as molecular markers for phylogenetic analyses [18,19]. Overall, these highly divergent regions presented abundant information for molecular marker development in plant identification and phylogenetic relationships of *Alpinia*. 

### 2.6. Interspecific Analyses of Alpinia Chloroplast Genomes

Using the *A. pumila* chloroplast genome sequence as the reference, we compared the SNP/indel loci of the other four chloroplast genomes in the current study. One-hundred-and-twenty-eight and 176 SNP markers were detected between *A. pumila* and *A. katsumadai* in protein-coding genes and non-coding regions, respectively (Appendix A). One-hundred-and-sixty-two and 205 SNP markers were detected between *A. pumila* and *A. oxyphylla* sampled from Guangdong in protein-coding genes and non-coding regions, respectively (Appendix A). One-hundred-and-forty-two and 189 SNP markers were detected between *A. pumila* and *A. zerumbet* in protein-coding genes and non-coding regions, respectively (Appendix A). One-hundred-and-sixty-three and 208 SNP markers were detected between *A. pumila* and *A. oxyphylla* sampled from Hainan in protein-coding genes and non-coding regions, respectively (Appendix A). The SNPs in the *A. katsumadai* chloroplast genome were significantly fewer than those in the two accessions of *A. oxyphylla*. SNPs were detected in 54 protein-coding genes in *A. katsumadai*, *A. zerumbet*, and two accessions of *A. oxyphylla* sampled from Guangdong and Hainan. Nine genes were in the SSC region, one gene was in the IR regions, and 44 genes were in the LSC region (Table 3 and Appendix A). These 54 genes were divided into four categories according to their different functions in plant chloroplasts, including photosynthetic apparatus, photosynthetic metabolism, gene expression, and other genes (Table 2). For the 162 and 163 SNP markers in the protein-coding genes of *A. oxyphylla* sampled from Guangdong and Hainan chloroplast genomes, respectively, 90 and 91 belonged to the synonymous type, and 72 and 72 belonged to the nonsynonymous type (Table 3 and Appendix A). Synonymous and nonsynonymous SNP markers in the protein-coding genes shared very similar number in these two chloroplast genomes. There were 67 synonymous SNPs and 61 nonsynonymous SNPs in the protein-coding genes of the *A. katsumadai* chloroplast genome (Table 3 and Appendix A). Seventy-three synonymous and 69 nonsynonymous SNP sites were detected in the chloroplast genome of *A. zerumbet* (Table 3 and Appendix A).

All of the indels were classified into insertions and deletions (Figure 8 and Appendix A). Fifty-six insertions and 62 deletions were detected between *A. pumila* and *A. katsumadai* chloroplast genomes, respectively (Figure 8A–C). Fifty-three insertions and 69 deletions were detected between *A. pumila* and *A. oxyphylla* sampled from Guangdong chloroplast genomes, respectively (Figure 8A–C). Sixty-five insertions and 50 deletions were detected between *A. pumila* and *A. zerumbet* chloroplast genomes, respectively (Figure 8A–C). Fifty-three insertions and 67 deletions were detected between *A. pumila* and *A. oxyphylla* sampled from Hainan chloroplast genomes, respectively (Figure 8A–C). Eight protein-coding genes from the four *Alpinia* accessions contained indels (Figure 8D). The gene *rpoC2* was a hotspot for indel variation, and all the four *Alpinia* accessions contained two indels in this gene. Comparison with two *Kaempferia* species was extremely interesting. The result indicated that SNPs between the four *Alpinia* species were less common than those between two *Kaempferia* species, but had more indels than those between two *Kaempferia* species. There were 536 SNPs and 107 indels between *K. galanga* and *K. elegans* [23]. Moreover, the SNPs obtained from chloroplast genomes had been successfully used for phylogenetic studies in several Zingiberaceae species, such as in *K. galanga* and *K. elegans* [23], *S. involucratus* [31], *H. coronarium* [32] and *C. longa* [33]. Therefore, these SNPs and indels in *Alpinia* here would be potential genetic markers to facilitate phylogenetic analysis and species identification in the family Zingiberaceae. 

### 2.7. Intraspecific Analyses of two Chloroplast Genomes of A. oxyphylla 

The two chloroplast genomes from *A. oxyphylla* were found to show a 59-bp difference in length (Table 1). In addition to the total length difference, we obtained indels and SNPs between the two chloroplast genomes of *A. oxyphylla* in their entirety. Through intraspecific comparison, a total of 23 indels were identified between the two A. oxyphylla accessions (Appendix A). Two, one, one, one, and one indels were located in *clpP*, *rpoC1*, *rps16*, *rpl16* and *trnG-GCC*, respectively. The other 17 indels were located in 14 different regions. *atpB*-*rbcL*, *trnC-GCA-petN* and *ndhF-rpl32* exhibited the same number indels, of two in total, respectively. There were 20 SNPs identified in the two chloroplast genomes of *A. oxyphylla* (Appendix A). The most frequently occurring mutations were A/C substitutions (4 times), followed by G/T (three times), T/A (three times), and T/G (three times), respectively. *ccsA-ndhD* contained the highest number of SNPs (6), followed by *rps16-trnQ*-*UUG* and *trnS-GCU*-*trnG-GCC*, each of which showed three SNPs. Two and two SNPs were located in introns and coding regions, respectively. All of the other four regions contained only one SNP. By contrast, there were more mutational events (SNPs and indels) in *Eucommia ulmoides* [47] and *Scutellaria baicalensis* [48]. There were 75 SNPs and 80 indels in two different individuals of *E. ulmoides* [47], 25 SNPs and 29 indels between the two *S. baicalensis* genotypes [48]. These 23 indels and 20 SNPs could be used for identification of different sources of *A. oxyphylla*.

### 2.8. Phylogenetic Analyses

To determine the phylogenetic relationships of five *Alpinia* accessions in family Zingiberaceae, two phylogenetic trees of 25 representatives from family Zingiberaceae using chloroplast SNP-based matrix, were constructed with *Costus pulverulentus*, *Costus viridis* and *Canna indica* as outgroups (Figure 9 and Appendix A). Both the maximum likelihood (ML) and maximum parsimony (MP) phylogenetic trees strongly indicated that two genera *Amomum* and *Alpinia* in the Zingiberaceae clade were strongly supported as a sister monophyletic group, respectively (bootstrap values ≥ 99%). In the genus *Alpinia*, two accessions of *A. oxyphylla* clustered together with high statistical support values (bootstrap values ≥ 99%), which was subsequently sister to *A. pumila*; *A. zerumbet* and *A. katsumadai* clustered together (bootstrap values ≥ 99%) was another sister to *A. pumila* (Figure 9 and Appendix A). Both ML and MP trees confirmed the previously known phylogenetic relationships in the family Zingiberaceae based on earlier studies [12,20,23,24,25,31,32,33,34], while unexpected relationships and positions of certain taxa were also revealed in this study. The reconfirmation of previously known relationships included (1) the monophyletic genera of *Amomum*, *Alpinia*, *Kaempferia*, *Zingiber* and *Hedychium* and their relationships to the rest of family Zingiberaceae, (2) the paraphyletic genus *Curcuma* in family Zingiberaceae, (3) relationships between two genera of *Curcuma* and *Stahlianthus*. The most surprising and unexpected findings in this study were the positions and relationships of *A. zerumbet* and *A. oxyphylla*. The two accessions of *A. oxyphylla* formed a group both in the two phylogenetic trees, confirming the validity of the assembled and annotated chloroplast genome of *A. oxyphylla* in this study. On morphological classifications, *A. pumila* was placed in subgenus *Alpinia*, *A. zerumbet* and *A. katsumadai* were placed in subgenus *Catimbium*, and *A. oxyphylla* was placed in subgenus *Probolocalyx* [11]. Therefore, our phylogenetic results were congruent with the morphological taxa [11]. On the other hand, several previous studies suggested that, *A. oxyphylla* was closely related to *A. zerumbet* based on chloroplast genomes [13], four groups were identified in genus *Alpinia* based on the *ITS* and *matK* trees [12]. We analyzed the reasons for the incongruence between chloroplast SNP-based phylogenetic analyses and chloroplast genomes, *ITS* and *matK* phylogenies in the following. Firstly, we did not have the same species samples as the *ITS* and *matK* phylogenies [12]. Secondly, we detected further phylogenetic relationships between *A. zerumbet* and *A. oxyphylla* with more chloroplast genomes data than the previous study [13]. Thirdly, with more than 230 species, the positions of *Alpinia* in family Zingiberaceae still remained somewhat uncertain and the future phylogenetic analyses should obtain additional chloroplast genomes of *Alpinia* species. The current phylogenetic analyses could supply the possibility that chloroplast genomes may be useful for phylogeny and species identification in family Zingiberaceae in the future. 

## 3. Materials and Methods

### 3.1. Plant Material and DNA Extraction

Fresh leaves were obtained from *A. katsumadai* (voucher specimen: Ak 2015001), *A. oxyphylla* sampled from Guangdong (voucher specimen: Ao_Liu 2015002) and *A. pumila* (voucher specimen: Ap_Liu 2015003) plants, respectively, from the resource garden of the Environmental Horticulture Research Institute, Guangdong Academy of Agricultural Sciences, Guangzhou, China. The total chloroplast DNA was extracted from those leaves using the modified sucrose gradient centrifugation method [49]. The chloroplast DNA sample of good integrity and with both optical density (OD) 260/280 and OD 260/230 ratio greater than 1.8 was used for subsequent experiments. 

### 3.2. Chloroplast Genome Sequencing and Assembly

For each sample, two libraries with insert sizes of 300 bp and 10 kb were constructed after purification, and then sequenced on an Illumina Hiseq X Ten instrument (Biozeron, Shanghai, China) and a PacBio Sequel platform (Biozeron, Shanghai, China), respectively. The resulting Illumina raw data and PacBio raw data were assessed with FastQC. A total of 66.3 M, 80.2 M and 66.4 M clean data of 150 bp Illumina paired-end reads were generated from *A. katsumadai*, *A. oxyphylla* sampled from Guangdong and *A. pumila*, respectively, and 0.62M, 0.56 M and 0.50 M clean data of 8–10 kb subreads were generated from the three species, respectively. 

Firstly, the Illumina paired-end clean reads were assembled using SOAPdenova (version 2.04) with default parameters into principal contigs [50], and all contigs were sorted and joined into a single draft sequence using the software Geneious version 11.0.4 [51]. Next, BLASR software was used to compare the PacBio clean data with the single draft sequence and to extract the correction and error correction [52]. Next, the corrected PacBio clean data were assembled using Celera Assembler (version 8.0) with default parameters, thus generating scaffolds [53]. Next, the assembled scaffolds were mapped back to the Illumina clean reads using GapCloser (version 1.12) for gap closing [50]. Finally, the redundant fragments sequences were removed, thus generating the final assembled chloroplast genomic sequence.

### 3.3. Chloroplast Genome Annotation and Structure Analysis

The assembled chloroplast genome was annotated using the online tool DOGMA (Dual Organellar Genome Annotator) [54] with default parameters and then checked manually. BLASTn searches in the NCBI website were used to identify and confirm both tRNA and rRNA genes. Lastly, further verification of the tRNA genes was carried out using tRNAscanSE with default settings [55]. The circular physical map of the chloroplast genome was drawn using OGDRAWv1.3.1 program with default parameters and subsequent manual editing [56]. The GenBank accession numbers of *A. katsumadai*, *A. oxyphylla* sampled from Guangdong and *A. pumila* are MK262728, MK262729 and MK262731, respectively. 

SSRs were identified using MISA (http://pgrc.ipk-gatersleben.de/misa/) [57]. The parameters for SSRs were adjusted for identification of perfect mono-, di-, tri-, tetra-, pena-, and hexanucleotide motifs with a minimum of 8, 5, 4, 3, 3, and 3 repeats, respectively. Furthermore, the size and location of long repeat sequences, including forward, palindrome, reverse and complement repeats, were determined by the online software REPuter [58], with a hamming distance of 3 and a mininal repeat size of 30 bp. 

### 3.4. Codon Usage and Prediction of RNA Editing Sites

Codon usage was determined for all protein-coding genes. To examine the deviation in synonymous codon usage, the relative synonymous codon usage (RSCU) was calculated using MEGA7 software [59]. Amino acid frequency was also calculated and expressed by the percentage of the codons encoding the same amino acid divided by the total codons. To predict the possible RNA editing sites in the three *Alpinia* species chloroplast genomes, protein-coding genes were used to predict potential RNA editing sites using the online program Predictive RNA Editor for Plants (PREP) suite (http://prep.unl.edu/) with a cutoff value of 0.8 [60]. 

### 3.5. Genome Comparison and Divergence Analyses

To compare the chloroplast genome of *A. pumila* with other four *Alpinia* accessions (*A. katsumadai*, *A.zerumbet*, and two accessions of *A. oxyphylla* sampled from Guangdong and Hainan), the mVISTA program (http://genome.lbl.gov/vista/mvista/about.shtml) in the Shuffle-LAGAN mode [61] was carried out using the annotation of *A. pumila* as the reference. To detect the variation in the LSC/IR/SSC boundaries of *Alpinia* chloroplast genomes, five *Alpinia* whole genomes were included in comparisons. The nucleotide variability (Pi) among chloroplast genomes was performed using software DnaSP v5 [62], with window length of 600 bp and the step size of 200 bp. The five *Alpinia* whole genomes were also aligned using MUMmer software [63] and adjusted manually where necessary using Se-Al 2.0 [64], using the annotated *A. pumila* chloroplast genome as the reference. The SNPs and indels were recorded separately as well as their locations in the chloroplast genome. The two accessions of *A. oxyphylla* were analyzed to identify SNPs and indels markers that can be selected in subsequent different sources studies. 

### 3.6. Phylogenetic Analyses

In this study, a total of 25 complete chloroplast genome sequences were downloaded from the GenBank (NCBI) database. *C. pulverulentus*, *C. viridis* and *C. indica* were used as outgroups of the family Zingiberaceae. Phylogenetic trees were constructed using SNP matrix from 25 representative chloroplast genomes of the family Zingiberaceae. The reliable SNP sites were obtained by previously described method [23]. For each chloroplast genome, all SNP sites were connected in the same order to obtain a sequence in FASTA format. Multiple FASTA format sequences alignments were carried out by ClustalW in software MEGA7 [59]. To examine the phylogenetic applications of rapidly evolving 11,112 SNP markers (Appendix A), maximum likelihood (ML) analysis based on the nucleotide substitution model of Tamura-Nei was conducted using software MEGA7 [59]. The maximum parsimony (MP) method was also employed to construct a phylogenetic tree using the Close-Neighbor-Interchange (CNI) model in software MEGA7 [59]. Both ML and MP analyses were performed with 1000 bootstrap replicates, respectively.

## 4. Conclusions

In summary, we sequenced and characterized the complete chloroplast genomes of three *Alpinia* (*A. katsumadai*, *A. pumila* and *A. oxyphylla* sampled from Guangdong) of the family Zingiberaceae. Then, we compared them to the two reported chloroplast genomes of *A. zerumbet* and *A. oxyphylla* sampled from Hainan. These five *Alpinia* chloroplast genomes were highly conserved in terms of gene order, GC content, SSRs, and long repeats, and were typical quadripartite circle molecules consisting of an LSC region, an SSC region, and a pair of separated IRs. The IR expansions and contractions of the border regions resulted in difference of genome size between five *Alpinia* accessions. Fifteen highly divergent regions (*rpl36*, *ycf1*, *rps15*, *rpl22*, *infA*, *psbT-psbN*, *accD-psaI*, *petD-rpoA*, *psaC-ndhE*, *ccsA-ndhD*, *ndhF-rpl32*, *rps11-rpl36*, *infA-rps8*, *psbC-psbZ*, and *rpl32-ccsA*) were found and could be used as potential markers for *Alpinia* species on plant identification and phylogenetic studies. Additionally, for the interspecific comparisons, 304, 367, 331 and 371 SNPs were detected in comparisons of *A. pumila-A. katsumadai*, *A. pumila-A. oxyphylla* (Guangdong), *A. pumila-A. zerumbet*, and *A. pumila-A. oxyphylla* (Hainan), respectively, when the *A. pumila* chloroplast genome was used as the reference; 118, 122, 115, and 120 indels were accurately located in these four comparisons, respectively. Through the intraspecific comparison, 20 SNPs and 23 indels were identified in the two accessions of *A. oxyphylla*. The ML and MP trees indicated that the chloroplast SNP-based phylogenetic analyses could be used to identify the five *Alpinia* accessions. *Alpinia* showed a close phylogenetic relationship with *Amomum* species. The molecular data in this study represent a valuable resource for the studies of phylogenetic relationships and species identification in the family Zingiberaceae.

## Figures and Tables

**Figure 1 plants-09-00286-f001:**
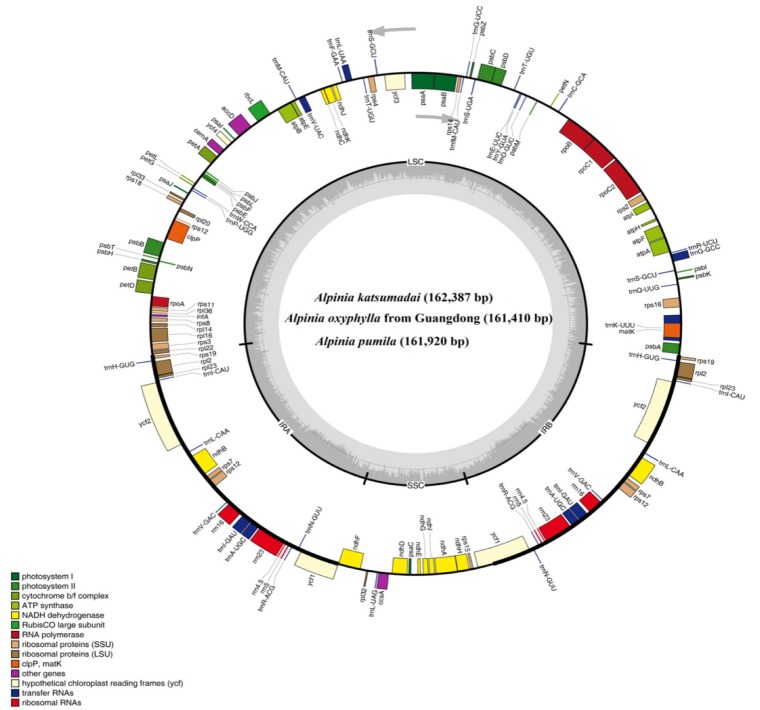
Circular gene map of chloroplast genomes of three *Alpinia* species. The gray arrowheads indicate the direction of the genes. Genes shown inside the circle are transcribed clockwise and those outside are transcribed counterclockwise. Different genes are color coded. The innermost darker gray corresponds to GC content, whereas the lighter gray corresponds to AT content. IR, inverted repeat; LSC, large single copy region; SSC, small single copy region.

**Figure 2 plants-09-00286-f002:**
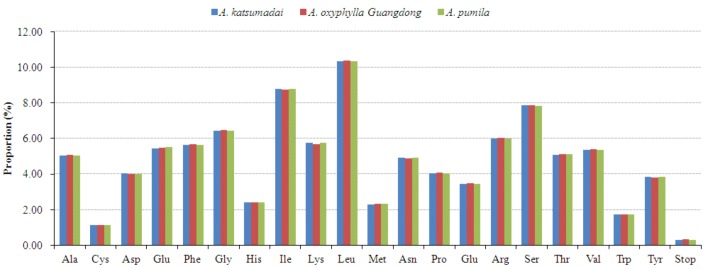
Amino acid proportion in three *Alpinia* species protein-coding sequences.

**Figure 3 plants-09-00286-f003:**
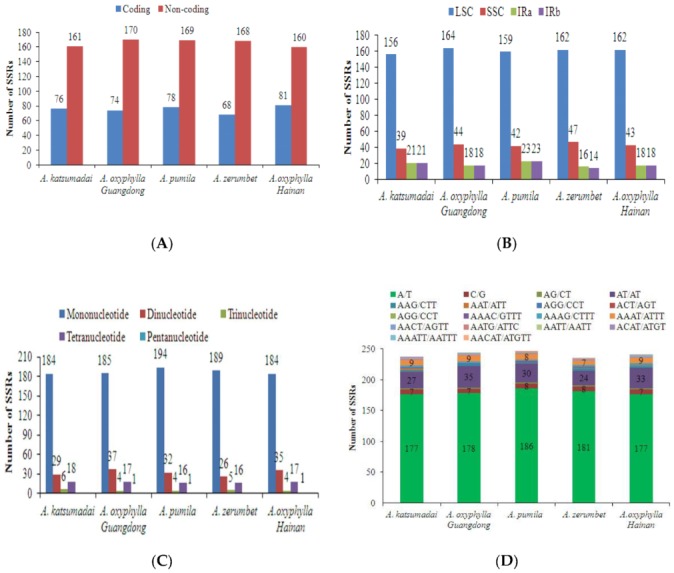
Distribution of small sequence repeats (SSRs) in the chloroplast genomes of five *Alpinia* accessions. (**A**) SSR distribution between coding and non-coding regions detected in five *Alpinia* chloroplast genomes; (**B**) Frequencies of identified SSRs in the large single copy (LSC), SSC and inverted repeats (IRs) regions; (**C**) Number of different SSR types detected in five *Alpinia* chloroplast genomes; (**D**) Frequency of identified SSRs in different repeat class types.

**Figure 4 plants-09-00286-f004:**
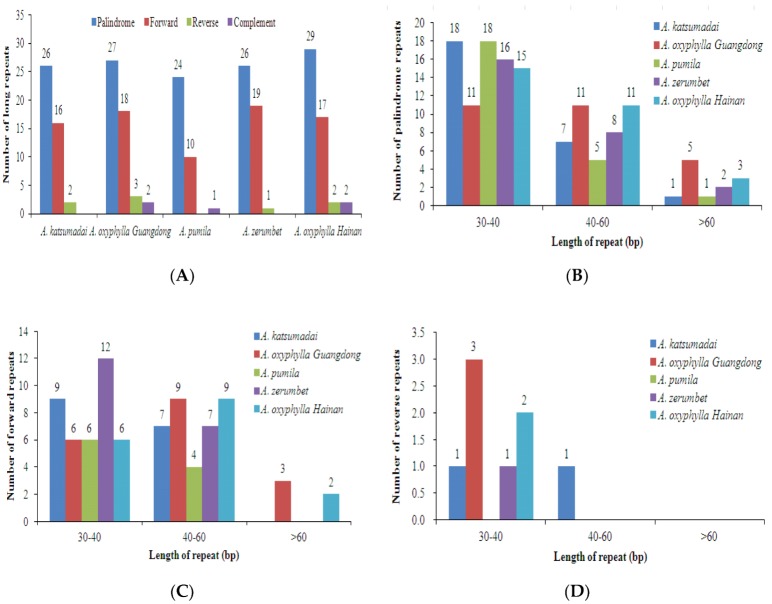
Analysis of long repeat sequences in the five *Alpinia* chloroplast genomes. (**A**) Total of four long repeat types; (**B**) Frequency of the palindromic repeats by length; (**C**) Frequency of the forward repeats by length; (**D**) Frequency of the reverse repeats by length.

**Figure 5 plants-09-00286-f005:**
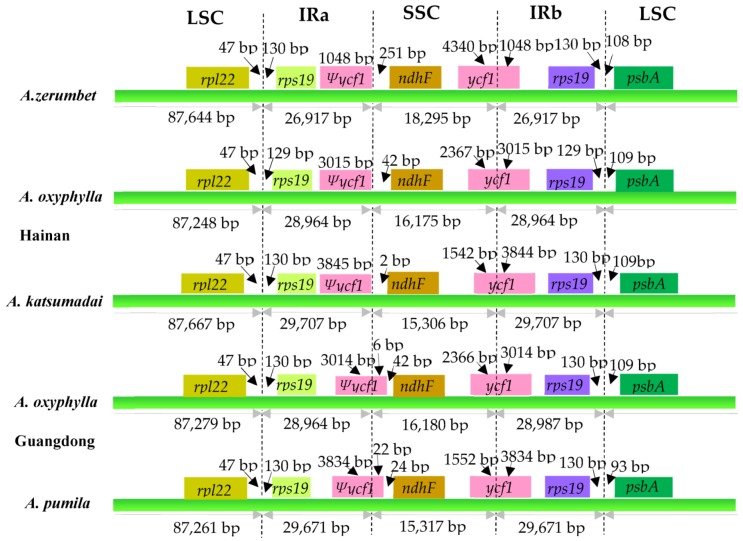
Comparison of the borders of the LSC, SSC, and IR regions among five *Alpinia* chloroplast genomes. Ψ, pseudogenes. Boxes above the main line indicate the adjacent border genes. The figure is not to scale with respect to sequence length, and only shows relative changes at or near the IR/SC borders.

**Figure 6 plants-09-00286-f006:**
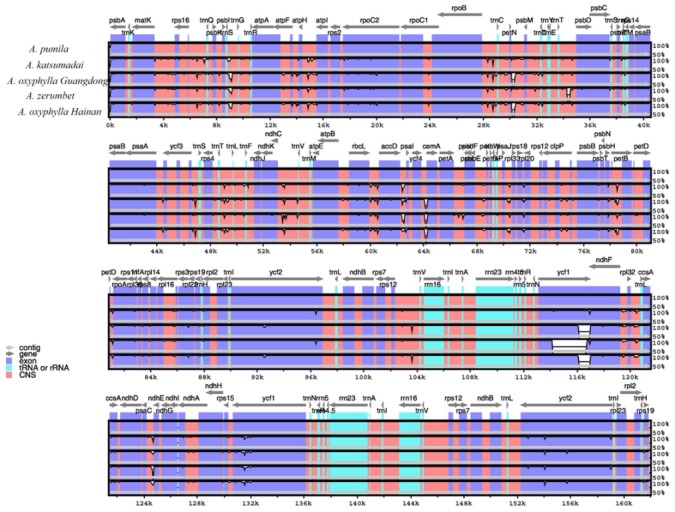
Complete chloroplast genome comparison of five *Alpinia* accessions using the chloroplast genome of *A. pumila* as a reference. Gray arrows and thick black lines above the alignment indicate gene orientation. Purple bars represent exons, sky-blue bars represent transfer RNA (tRNA) and ribosomal RNA (rRNA), red bars represent non-coding sequences (CNS), and white peaks represent differences of chloroplast genomes. The y-axis represents the identity percentage ranging from 50 to 100%.

**Figure 7 plants-09-00286-f007:**
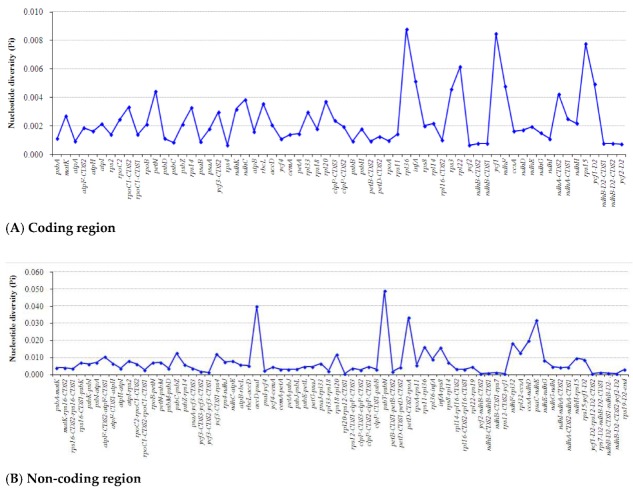
Sliding window analysis of the whole chloroplast genomes among five *Alpinia* accessions.Window length: 800 bp; step size: 200 bp. X-axis: position of the midpoint of a window.

**Figure 8 plants-09-00286-f008:**
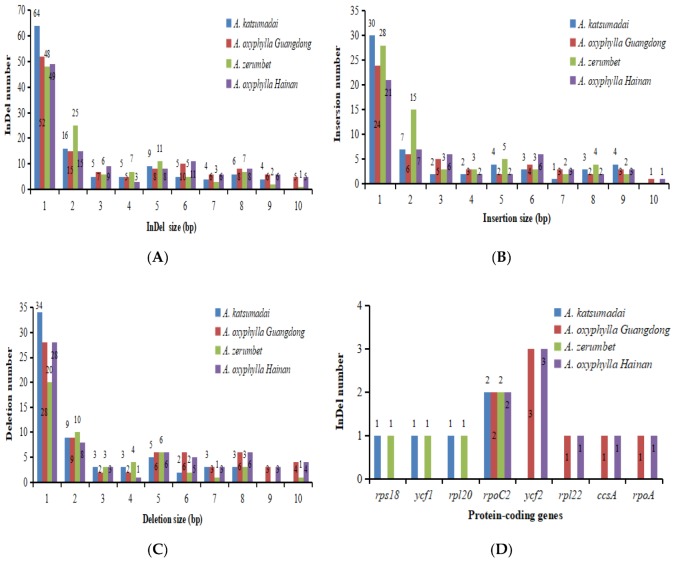
Indels statistics of four *Alpinia* chloroplast genomes. The *A. pumila* chloroplast genome was used as the reference sequence for indels analyses for the other four chloroplast genomes. (**A**) Total indels statistics. (**B**) Insertion statistics. (**C**) Deletion statistics. (**D**) Indels belonging to different protein-coding genes.

**Figure 9 plants-09-00286-f009:**
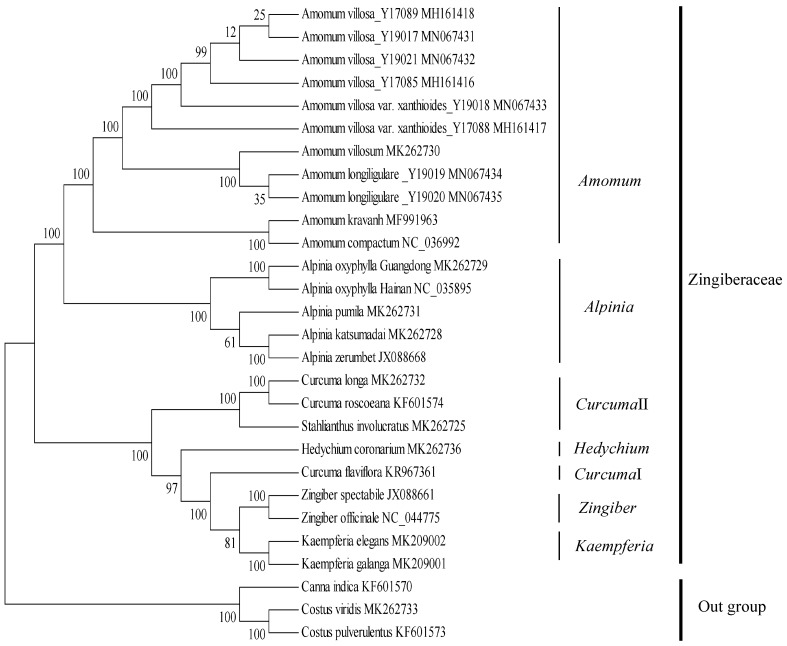
Phylogenetic tree constructed with SNPs from 28 chloroplast genomes using maximum likelihood method. The bootstrap values were based on 1000 replicates and were indicated next to the branches.

**Table 1 plants-09-00286-t001:** Comparison of the general features of the five *Alpinia* chloroplast genomes.

Genome Characteristics	*A. katsumadai*	*A. oxyphylla* Guangdong	*A. pumila*	*A. zerumbet*	*A. oxyphylla* Hainan
GenBank number	MK262728	MK262729	MK262731	JX088668	NC_035895
Genome size (bp)	162,387	161,410	161,920	159,773	161,351
LSC length (bp)	87,667	87,279	87,261	87,644	87,248
SSC length (bp)	15,306	16,180	15,317	18,295	16,175
IR length (bp)	29,707	28,964/28,987	29,671	26,917	28,964
Total genes	133	133	133	132	132
Protein-coding genes	87	87	87	86	87
tRNA genes	38	38	38	38	37
rRNA genes	8	8	8	8	8
GC content (%)	36.15	36.16	36.17	36.27	36.17

**Table 2 plants-09-00286-t002:** Genes present in the chloroplast genomes of three *Alpinia* species.

Category	Gene Names	Amount
Photosystem Ⅰ	*psaA*, *psaB*, *psaC*, *psaI*, *psaJ*	5
Photosystem Ⅱ	*psbA*, *psbB*, *psbC*, *psbD*, *psbE*, *psbF*, *psbH*, *psbI*, *psbJ*, *psbK*, *psbL*, *psbM*, *psbN*, *psbT*, *psbZ*	15
Cytochrome b/f complex	*petA*, *petB* *, *petD* *, *petG*, *petL*, *petN*	6
ATP synthase	*atpA*, *atpB*, *atpE*, *atpF* *, *atpH*, *atpI*	6
NADH dehydrogenase	*ndhA* *, *ndhB*(×2) *, *ndhC*, *ndhD*, *ndhE*, *ndhF*, *ndhG*, *ndhH*, *ndhI*, *ndhJ*, *ndhK*	12
Rubisco	*rbcL*	1
RNA polymerase	*rpoA*, *rpoB*, *rpoC1* *, *rpoC2*	4
Large subunit ribosomal proteins	*rpl2*(×2) *, *rpl14*, *rpl16* *, *rpl20*, *rpl22*, *rpl23*(×2), *rpl32*, *rpl33*, *rpl36*	11
Small subunit ribosomal proteins	*rps2*, *rps3*, *rps4*, *rps7*(×2), *rps8*, *rps11*, *rps12*(×2) *, *rps14*, *rps15*, *rps16* *, *rps18*, *rps19*(×2)	15
Other proteins	*accD, ccsA*, *cemA*, *clpP* **, *infA*, *matK*	6
Proteins of unknown function	*ycf1*(×2), *ycf2*(×2), *ycf3* **, *ycf4*	6
Ribosomal RNAs	*rrn4.5*(×2), *rrn5*(×2), *rrn16*(×2), *rrn23*(×2)	8
Transfer RNAs	*trnA-UGC*(×2) *, *trnC-GCA*, *trnD-GUC*, *trnE-UUC*, *trnF-GAA*, *trnfM-CAU*, *trnG-GCC* *, *trnG-UCC*, *trnH-GUG*(×2), *trnI-CAU*(×2), *trnI-GAU*(×2) *, *trnK-UUU*(×2) *, *trnL-CAA*(×2), *trnL-UAA* *, *trnL-UAG*, *trnM-CAU*, *trnN-GUU*(×2), *trnP-UGG*, *trnQ-UUG*, *trnR-ACG*(×2), *trnR-UCU*, *trnS-GCU*(×2), *trnS-UGA*, *trnT-UGU*, *trnV-GAC*(×2), *trnV-UAC* *, *trnW-CCA*, *trnY-GUA*	38

* Gene containing one intron; ** gene containing two introns; (×2) gene with two copies.

**Table 3 plants-09-00286-t003:** Comparisons of mutation changes, number of synonymous (S) and nonsynonymous (N) substitutions per gene of protein-coding genes among four *Alpinia* chloroplast genomes.

Genes	*A. katsumadai*		*A. oxyphylla* Guangdong		*A. zerumbet*		*A. oxyphylla* Hainan		Location
	S	N	S	N	S	N	S	N	
*psbA*	3	0	3	0	3	0	3	0	LSC
*matK*	2	1	3	3	2	2	3	3	LSC
*atpA*	1	1	2	1	1	1	2	1	LSC
*atpF*	-	-	-	-	2	0	-	-	LSC
*atpH*	1	0	1	0	1	0	1	0	LSC
*atpI*	1	1	-	-	2	1	-	-	LSC
*rps2*	1	0	1	1	1	0	1	1	LSC
*rpoC2*	5	7	8	8	6	7	8	8	LSC
*rpoC1*	0	2	6	3	0	2	6	3	LSC
*rpoB*	7	2	7	3	6	3	7	3	LSC
*petN*	-	-	-	-	1	0	-	-	LSC
*psbD*	-	-	-	-	1	2	-	-	LSC
*psbC*	1	0	0	1	1	0	0	1	LSC
*psbZ*	1	0	1	0	1	0	1	0	LSC
*rps14*	-	-	0	1	1	0	0	1	LSC
*psaB*	2	0	2	1	1	0	2	1	LSC
*psaA*	6	1	4	1	5	1	4	1	LSC
*ycf3*	-	-	1	0	-	-	1	0	LSC
*rps4*	1	0	1	0	1	0	1	0	LSC
*ndhK*	0	2	1	1	1	3	1	1	LSC
*ndhC*	1	1	0	2	0	1	0	2	LSC
*atpB*	-	-	2	2	1	1	3	2	LSC
*rbcL*	3	4	5	1	3	3	5	1	LSC
*accD*	2	1	3	2	2	1	3	2	LSC
*ycf4*	-	-	1	0	-	-	1	0	LSC
*cemA*	1	0	1	1	1	0	1	1	LSC
*petA*	0	2	1	2	0	2	1	2	LSC
*rpl33*	-	-	0	1	-	-	0	1	LSC
*rps18*	-	-	1	0	-	-	1	0	LSC
*rpl20*	-	-	2	0	1	0	2	0	LSC
*clpP*	1	0	1	0	1	0	1	0	LSC
*psbB*	1	0	-	-	3	0	-	-	LSC
*psbH*	1	0	1	0	1	0	1	0	LSC
*petB*	1	0	-	-	1	0	-	-	LSC
*petD*	-	-	0	1	-	-	0	1	LSC
*rpoA*	1	0	0	1	-	-	0	1	LSC
*rps11*	-	-	0	1	-	-	0	1	LSC
*rpl36*	1	0	2	0	1	0	2	0	LSC
*infA*	1	0	1	0	1	0	1	0	LSC
*rps8*	2	0	2	0	2	0	2	0	LSC
*rpl14*	1	0	1	0	1	1	1	0	LSC
*rpl16*	-	-	-	-	0	1	-	-	LSC
*rps3*	3	2	2	3	2	3	2	3	LSC
*rpl22*	0	3	0	1	0	3	0	1	LSC
*ndhF*	5	7	8	5	6	6	8	5	SSC
*ccsA*	0	3	0	2	1	2	0	2	SSC
*ndhD*	-	-	3	0	-	-	3	0	SSC
*ndhE*	-	-	1	0	-	-	1	0	SSC
*ndhG*	2	0	1	0	1	0	1	0	SSC
*ndhI*	-	-	1	0	-	-	1	0	SSC
*ndhA*	2	2	2	1	2	2	2	1	SSC
*ndhH*	2	2	1	0	2	2	1	0	SSC
*rps15*	1	1	0	3	0	1	0	3	SSC
*ycf1*	2	15	6	19	3	18	6	19	IRa/b
Total	67	61	90	72	73	69	91	72	-

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
