# Peer review of "Complete Chloroplast Genomes of Three Medicinal Alpinia Species: Genome Organization, Comparative Analyses and Phylogenetic Relationships in Family Zingiberaceae"

_plants, 2020, doi:10.3390/plants9020286_

Round 1
Reviewer 1 Report
Dear authors,
You may reduce some keywords. Please edit the reference’s format. I find many errors. If the three medicinal plants species are from the gardens in the Guangdong academy of agricultural sciences, Guangzhou, you may obtain more plant profile in the horticulture, and then write those in the discussion. For example, the origin locations of the three plant materials. 42-43: Please edit those sentences. ……biodiversity of Alpinia 46,47,84 and so on, those citation numbers colors are not consistent. 47, I suggest that Alpinia pumila as Traditional Chinese folk medicine, not Traditional Chinese medicine. I only fine Alpinia katsumadai, Alpinia oxyphylla are Chinese medicine. Or you should give more robust reference to support the three plant species are “Traditional Chinese medicine”, for example, record in the The Compendium of Materia Medica (also known by the Bencao Gangmu) or in the Pharmacopoeia of the People's Republic of China (PPRC) or the Chinese Pharmacopoeia (ChP). 50, delete Chinese species. In taxonomy, you should write “ endemic to China” or “native to China”. 274-277, the word sizes are inconsistent. 400-408: rewrite Figure 9, higher resolution should be provided. 427, 452-454, I suggest to use MEGA7 or MEGA X because MEGA6 is really too old software version for 2019 and 2020 papers. Please write some “discussion” about your results of oxyphylla sampled from Hainan and Guangdong. For example, what it does mean for systematics based on your results? Why do you want to compare this species that resource from two locations? What you anticipate to discover from your results? You should address more.
Author Response
You may reduce some keywords. Please edit the reference’s format. I find many errors. If the three medicinal plants species are from the gardens in the Guangdong academy of agricultural sciences, Guangzhou, you may obtain more plant profile in the horticulture, and then write those in the discussion. For example, the origin locations of the three plant materials. 42-43: Please edit those sentences. ……biodiversity of Alpinia 46,47,84 and so on, those citation numbers colors are not consistent.
Response: Yes, we revised this places in the text.
47, I suggest that Alpinia pumila as Traditional Chinese folk medicine, not Traditional Chinese medicine. I only fine Alpinia katsumadai, Alpinia oxyphylla are Chinese medicine. Or you should give more robust reference to support the three plant species are “Traditional Chinese medicine”, for example, record in the The Compendium of Materia Medica (also known by the Bencao Gangmu) or in the Pharmacopoeia of the People's Republic of China (PPRC) or the Chinese Pharmacopoeia (ChP).
Response: We found the book (the Bencao Gangmu) again, and that Alpinia katsumadai and Alpinia oxyphylla are traditional Chinese medicine. However, in other books (such as ZHONGGUO YAOYONG JIANGKE ZHIWU, Zhongguo Yaoyong Zhiwu Zhi volume 12), Alpinia pumila has traditional practice use. Therefore, we agreed with this idea that Alpinia pumila is Traditional Chinese folk medicine. We revised in the text as following:
Of these species, the dried seeds of A. katsumadai and A. oxyphylla, and dried rhizomes of A. pumila can be used as traditional Chinese medicine and folk medicine, respectively.
50, delete Chinese species. In taxonomy, you should write “ endemic to China” or “native to China”. 274-277, the word sizes are inconsistent.
Response: we agreed with this idea and deleted “Chinese species”.
400-408: rewrite Figure 9, higher resolution should be provided.
Response: Yes, we also found Figure 9 had low resolution in the pdf form of the MS. We supplied the original Figure 9 with higher resolution.
427, 452-454, I suggest to use MEGA7 or MEGA X because MEGA6 is really too old software version for 2019 and 2020 papers.
Response: we agreed with idea. We update the reference in the text and use the version 7 for phylogenetic tree analysis. The two phylogenetic trees are also updated.
Kumar, S; Stecher, G.; Tamura, K. Mega7: molecular evolutionary genetics analysis version 7.0 for bigger datasets. Mol. Biol. Evol. 2016, 33, 1870-1874.
Please write some “discussion” about your results of Alpinia oxyphylla sampled from Hainan and Guangdong. For example, what it does mean for systematics based on your results? Why do you want to compare this species that resource from two locations? What you anticipate to discover from your results? You should address more.
Response: Firstly, we did our works and wrote the text without Alpinia oxyphylla sampled from Hainan before Sep. 2019. But after Sep. 2019, we retrieved the NCBI database and found that the chloroplast genome of Alpinia oxyphylla sampled from Hainan was released to the world’s researchers. Then we downloaded the genome sequence from the NCBI database and analyzed the published chloroplast genomes of Alpinia zerumbet (A. zerumbet ) and A. oxyphylla sampled from Hainan to retrieve useful chloroplast molecular resources for Alpinia. Through the intraspecific comparison, 20 SNPs and 23 indels were identified in the two accessions of A. oxyphylla sampled from Guangdong and Hainan. Due to funding constraints, we did not collect more different resources of A. oxyphylla to verify our discoveries.

Reviewer 2 Report
Li and colleagues reported three chloroplast genomes from the genus Alpinia (A. katsumadai, A. oxyphylla, and A. pumila), and performed comparative analyses using five chloroplast genomes from this genus. The authors described the characteristics of their sequenced genomes in detail. One of their main findings is the existence of interspecific and intraspecific SNPs that could be used for molecular classification. The manuscript is well structured and would be acceptable for this journal after several modifications. I listed my suggestions and minor changes/corrections as follows. 1) the abstract seems to be too descriptive, and it should be shorten (the maximum is ~200 words). 2) page 2, line 66. “to resolve morphological discrepancies” would be “to resolve molecular phylogeny”. 3) page 5, line160. “A total of 56 editing sites in 22 protein-coding genes were identified in A. katsumadai”. This sentence is confusing, and the authors should add an explanation that all RNA editing sites were predicted only by in silico analysis. 4) page 15, line 354. “which was subsequently sister to A. pumila;….” will be modified to “which was subsequently sister to the clade consisting of A. pumila, A. zeerumbet, and A. katsumadai;…”. 5) page 15, line 360. “the paraphyletic genus” will not be italicized. 6) page 15, line 361. “The most surprising and unexpected findings in this study were the positions and relationships of A. zerumbet and A. oxyphylla.”. Please explain why this relationship was unexpected? 7) page 15, line 368. “A. oxyphylla was closely related to A. zerumbet based on chloroplast genomes [13]”. That is totally obvious because the earlier study used only these two species from the genus Alpinia. Therefore, the phylogeny of this present study is NOT conflict with that of the previous study. 8) page 15, line 370. “We analyzed the reasons for the incongruence between chloroplast genome SNP arrays and chloroplast genomes, ITS and matK phylogenies in the following.”. To discuss the reason, I highly recommend that the authors perform a phyletic analysis using ITS and matK sequences from 28 species used in this study, and compare tree topologies between the SNP-based and ITS & matK-based phylogenies. 9) page 15 and 17. “SNP arrays” is a kind of “DNA microarrays”. The authors should avoid using this term. “chloroplast genome SNP arrays” would be “chloroplast SNP-based phylogenetic analyses”. 10) page 17, line 402. “BLASR” would be “BLAST”. 11) page 17-18. In the method of phylogenetic analyses, the authors should give the information for the number of site positions (nucleotides) in the alignment and selected phylogenetic model.Author Response
Li and colleagues reported three chloroplast genomes from the genus Alpinia (A. katsumadai, A. oxyphylla, and A. pumila), and performed comparative analyses using five chloroplast genomes from this genus. The authors described the characteristics of their sequenced genomes in detail. One of their main findings is the existence of interspecific and intraspecific SNPs that could be used for molecular classification. The manuscript is well structured and would be acceptable for this journal after several modifications. I listed my suggestions and minor changes/corrections as follows.
- the abstract seems to be too descriptive, and it should be shorten (the maximum is ~200 words).
Response: we revised some words in the abstract. We retained the main contents of the abstract.
2) page 2, line 66. “to resolve morphological discrepancies” would be “to resolve molecular phylogeny”.
Response: we revised these words. According the text, we revised these words to “ to resolve morphological differences”.
3) page 5, line160. “A total of 56 editing sites in 22 protein-coding genes were identified in A. katsumadai”. This sentence is confusing, and the authors should add an explanation that all RNA editing sites were predicted only by in silico analysis.
Response: We agreed this idea and wrote in the section of Materials and methods.
4) page 15, line 354. “which was subsequently sister to A. pumila;….” will be modified to “which was subsequently sister to the clade consisting of A. pumila, A. zeerumbet, and A. katsumadai;…”.
Response: we did not agree with this idea. Based on the results in current study, in genus Alpinia, we suggested that the five Alpinia accessions were divided into three groups, not two groups.
5) page 15, line 360. “the paraphyletic genus” will not be italicized.
Response: We agreed this idea. We revised these words in regular form.
- page 15, line 361. “The most surprising and unexpected findings in this study were the positions and relationships of zerumbetand A. oxyphylla.”. Please explain why this relationship was unexpected?
Response: Because based on previous study [13], A. oxyphylla was closely related to A. zerumbet based on chloroplast genomes. But in our phylogenies analyses, two accessions of A. oxyphylla clustered together with high statistical support values (bootstrap values≧99%), which was subsequently sister to A. pumila; A. zerumbet and A. katsumadai clustered together (bootstrap values≧99%) was another sister to A. pumila. Therefore, the relathinships of A. zerumbet and A. oxyphylla were unexpected.
- page 15, line 368. “ oxyphyllawas closely related to A. zerumbet based on chloroplast genomes [13]”. That is totally obvious because the earlier study used only these two species from the genus Alpinia. Therefore, the phylogeny of this present study is NOT conflict with that of the previous study.
Response: According to the previous study [13], A. oxyphylla was closely related to A. zerumbet based on chloroplast genomes. Based on these results, if added more species from the genus Alpinia, A. oxyphylla should be closely related to A. zerumbet. But our phylogenies analyses were not consistent with the previous study [13].
- page 15, line 370. “We analyzed the reasons for the incongruence between chloroplast genome SNP arrays and chloroplast genomes, ITS and matK phylogenies in the following.”. To discuss the reason, I highly recommend that the authors perform a phyletic analysis using ITS and matK sequences from 28 species used in this study, and compare tree topologies between the SNP-based and ITS & matK-based phylogenies.
Response: Sorry, the ITS sequceces from 28 species used in this study were not published all. However, we can use matK sequences from 28 species used in this study for phylogenies analyses (Figure S2). The matK phylogenies results were consistent with our SNP-based phylogenies, but matK phylogenies were with lower statistical support values than those of our SNP-based phylogenies.
On the other hand, our SNP sites contained the mutations in the whole chloroplast genomes, including matK genes. By comparison, we finally select the SNP-based phylogenies trees in the text.
- page 15 and 17. “SNP arrays” is a kind of “DNA microarrays”. The authors should avoid using this term. “chloroplast genome SNP arrays” would be “chloroplast SNP-based phylogenetic analyses”.
Response: Yes, we agreed this idea. We revised “chloroplast genome SNP arrays” to “chloroplast genome SNP -based phylogenetic analyses” in line 353. We also revised the “SNP arrays” to “SNP matrix”.
- page 17, line 402. “BLASR” would be “BLAST”.
Response: We disagree. In this place, we should use “BLASR”.
11) page 17-18. In the method of phylogenetic analyses, the authors should give the information for the number of site positions (nucleotides) in the alignment and selected phylogenetic model.
Response: We added the the number of site positions (11,112 nucleotides) in the text and selected phylogenetic model (Tamura-Nei nucleotide substitution model). The revised text was following:
Multiple FASTA format sequences alignments were carried out by ClustalW in software MEGA7 [59]. To examine the phylogenetic applications of rapidly evolving 11,112 SNP markers (Supplement.SNP matrix.data), maximum likelihood (ML) analysis based on the nucleotide substitution model of Tamura-Nei was conducted using software MEGA7 [59]. The maximum parsimony (MP) method was also employed to construct a phylogenetic tree using Close-Neighbor-Interchange (CNI) model in software MEGA7 [59]. Both ML and MP analyses were performed with 1,000 bootstrap replicates, respectively.
